# Geometric Error Analysis and Compensation in Spherical Generating Grinding of Hemispherical Shell Resonators

**DOI:** 10.3390/mi13091535

**Published:** 2022-09-16

**Authors:** Yu Wang, Chaoliang Guan, Yifan Dai, Shuai Xue

**Affiliations:** 1College of Intelligence Science, National University of Defense Technology, Changsha 410073, China; 2Laboratory of Science and Technology on Integrated Logistics Support, National University of Defense Technology, Changsha 410073, China; 3Hunan Key Laboratory of Ultra-Precision Machining Technology, Changsha 410073, China

**Keywords:** hemispherical shell resonator, generating grinding, tool setting, tool wear, error compensation

## Abstract

The geometric accuracy of a hemispherical shell resonator (HSR) affects the assembly accuracy and final performance of a hemispherical resonant gyroscope in many ways. During the precision grinding of a resonator, the tool-setting error and wear error affect the form and positional accuracy of the inner and outer spherical surfaces. In this study, a compensation method for generating grinding of the HSR is proposed to address this problem. The geometric errors of the inner and outer spherical surfaces are systemically analyzed and a geometric model of the tool setting and wheel wear is established for generating grinding of the HSR. According to this model, a mapping relationship between the wheel pose and size, form, and positional error of the HSR was proposed. Experiments regarding machining, on-machine measurements, and error compensation were performed using the mapping relationship. The results demonstrate that the proposed method can reduce the radius error of the inner and outer spherical surfaces from 10 μm to 1 μm, sphericity from 5 μm to 1.5 μm, and concentricity from 15 μm to 3 μm following grinding. The form and positional errors are simultaneously improved, verifying the effectiveness of the proposed method.

## 1. Introduction

The hemispherical resonant gyroscope is one of the most accurate, stable, and reliable inertial sensors with the longest lifetime. The gyroscope has broad application prospects for navigation in aerospace, aviation, vehicles, and ships [1,2]. The hemispherical shell resonator (HSR) anchored by the central rod is the core of the hemispherical resonant gyroscope. The form and positional accuracy of the inner and outer spherical surfaces of the HSR affect the assembly accuracy and frequency characteristics of the gyroscopes in many ways, thereby affecting the sensitivity accuracy of the angular velocity or angular displacement of the gyroscope [3,4].

The HSR is commonly composed of fused silica and is shaped as a hemispherical thin-walled shell with a central support rod, as shown in Figure 1; the diameter of the hemisphere is typically 15–40 mm and the wall thickness is 1–2 mm. Its relatively small size, complex structure, and it being significantly hard and brittle make it difficult to manufacture. Generating grinding with a cup wheel is a common high-efficiency, high-precision, spherical forming method [5,6], by which the one-time formation of the HSR single spherical surface, support rod, and chamfer can be completed with a single-direction feeding and tool posture [7]. Studies regarding the application of this method to HSR machining are limited. According to the existing applications, for example, microlens [8] and graphite ball [9] processing, it can be inferred that the method not only ensures the form and positional accuracy of these structures, but also simplifies the complexity of the machine tool to improve its dynamic stiffness. However, tool wear that develops during the grinding process may affect the inner and outer spherical dimensions and concentricity. Moreover, the tool setting and dimensions of the grinding wheel for spherical surface grinding by the generation method affect the dimensions, form, and position of the spherical surface.

Wheel wear causes geometric errors that develop as the grinding progresses. Wear is often reduced by the arc envelope grinding method (AEGM) [10], a method that applies multiple points on the grinding wheel in an arc to participate in the entire grinding process. In this manner, the wheel wear is dispersed to different parts of the front end of the wheel, while remaining to deviate the shape of the machined surface. An on-machine dressing method restores the geometric accuracy lost owing to the wheel wear by the electrical discharge machining of the grinding wheels [11]. In high-steep aspheric surface grinding with an arc envelope, a model and prediction of the wear distribution of the grinding wheel are proposed to change the tool path to provide compensation [12]. A form-truing error compensation approach is proposed using an approached wheel-arc profile to replace the previously designed ideal profile. The objective is to directly compensate for the wheel arc profile errors [13]. These methods are based on the point-contact grinding of the AEGM. The contact points on the grinding wheel profile change with different workpiece machining parts. For an HSR with small, complex structures, point contact grinding requires multidimensional machine tool motion to ensure the accuracy of the HSR spherical surface [14,15], and the grinding wheel wears quickly, requiring frequent monitoring and adjustment in the mid-process to ensure the certainty of the contact point [16]. Spherical grinding often utilizes a specific ring line of the cup wheel to simultaneously grind a wide range of spherical surfaces, effectively ensuring the rotational symmetry of the spherical surface around the workpiece axis and increasing the grinding efficiency [6,17].

The installation and reinstallation of the wheel causes tool-setting errors. A truing and dressing technique of the cup wheel using loose abrasives was introduced to compensate for the dimensions of the workpieces caused by the tool setting [7]. A numerical simulation method was used to establish the relationship between the wheel-setting error and grinding error. The method uses an approximate parameter model to fit the error to improve the grinding accuracy [18]. However, there are coupling terms in the parameter identification of this model, and the tool setting error cannot be completely solved. Therefore, the method cannot be applied to HSR, a workpiece that requires not only form, but also positional accuracy. Currently, the compensation of the tool setting error in spherical generation grinding is either a single object, or the method is complicated or limited, which is insufficient for simultaneously meeting the requirements of size, form, and positional accuracy.

Therefore, we reported a wheel adjustment compensation method by establishing the mapping relationship between the wheel pose and size, form, and positional error of the HSR. First, we theoretically analyzed the errors caused by the tool setting and wear in generating spherical grinding with a cup wheel. Moreover, we simplified the compensation strategy for the size, form, and positional error. Finally, experiments regarding the inner and outer spherical machining, on-machine measurement, and compensation were conducted to verify the compensation accuracy and effect.

## 2. Method

### 2.1. Analysis of Tool Setting Error on HSR Machining

The geometric model of the tool and workpiece during HSR grinding is shown in Figure 2, where *XYZ* denotes the machine tool coordinate system, *C* denotes the workpiece rotation axis, and *A* denotes the wheel rotation axis. The rotations of the two spindle and *X*/*Z* axis feed motions can complete the one-time formation of the outer/inner spherical surface, support rod, and chamfer. According to the characteristics of generating spherical grinding with a cup wheel, the relative position between the wheel and the workpiece axis directly determines the formation accuracy of the spherical surface, provided that the motion errors of the two rotating axes are neglected.

**Theorem** **1.**
*The wheel rotating axis and workpiece rotating axis must spatially intersect for the cup wheel to theoretically generate an ideal spherical surface.*


**Proof** **of** **Theorem 1.**The geometric model of the wheel involved in grinding is indicated by the torus, as shown in Figure 3. Considering the geometric center as the origin and the axis of rotational symmetry as the *Z_w_* axis, the coordinates of any point (*x_2_, y_2_, z_2_*) on the torus can be expressed as follows:(1)$${\left(\sqrt{x_{2}^{2}+y_{2}^{2}}-R\right)}^{2}+z_{{}_{2}}^{2}=r^{2}$$The corresponding domain function is as follows:(2)$$R_{t}-r\leq \sqrt{x_{2}^{2}+y_{2}^{2}}\leq R_{t}+r$$
where *R_t_* is the mean diameter of the torus and *r* is the radius of the arc of the torus end. The range represented by Equation (1) can be divided into the following two parts:(3)$$0\leq \sqrt{x_{2}^{2}+y_{2}^{2}}\leq R_{t}-r$$
(4)$$R_{t}-r\leq \sqrt{x_{2}^{2}+y_{2}^{2}}\leq 0$$When the wheel axis space intersects the workpiece axis, as shown in Figure 4, the pose and position of the wheel have five degrees of freedom: (*z*, *α*, *φ*, *d*, *θ*), where *z* is the distance translated along the *Z*-axis, *α* is the rotation around the *Y*-axis, *φ* is the rotation around the *Z*-axis, and *d* is the distance from the center point of the grinding wheel to the axis of the workpiece along the axis of the grinding wheel. The pose and position of the wheel relative to the workpiece axis can be considered as five transformations in that order. The sphere is formed by the envelope of the rotation of two axes; therefore, *δ_z_*, *φ*, and *θ* do not affect the form of the processed spherical surface. Namely, two degrees of freedom, *α* and *d*, affect the form accuracy of the spherical surface.First, we prove Theorem 1 when *α* = 90°. The torus has a translation *d* along the *Z* axis and a pose transformation of the rotation α = 90° around the *Y*-axis. Under this condition, the equation for the torus can be obtained as follows:(5)$${\left(\sqrt{\left(y_{1}^{2}+z_{1}^{2}\right)}-R\right)}^{2}+{\left(x_{1}-d\right)}^{2}=r^{2}$$Let $F={\left(\sqrt{y_{1}^{2}+z_{1}^{2}}-R\right)}^{2}+{\left(x_{1}-d\right)}^{2}-r^{2}$. The normal equation of any point $\left(x_{0},y_{0},z_{0}\right)$ on the torus can be expressed as follows:(6)$$\frac{x-x_{0}}{F_{x}\left(x_{0},y_{0},z_{0}\right)}=\frac{y-y_{0}}{F_{y}\left(x_{0},y_{0},z_{0}\right)}=\frac{z-z_{0}}{F_{z}\left(x_{0},y_{0},z_{0}\right)}$$Substituting (0, 0, 0) into Equation (6), and in conjunction with Equation (5), the points on the torus whose normal intersects the origin (0, 0, 0) can be expressed as follows:(7)$$\left\{\begin{array}{l} \sqrt{y_{0}^{2}+z_{0}^{2}}=R_{t}\pm \frac{R_{t}r}{\sqrt{R_{t}^{2}+d^{2}}}\\ x_{0}=d\pm \frac{dr}{\sqrt{R_{t}^{2}+d^{2}}} \end{array}\right.$$These points apparently satisfy the following:(8)$$\sqrt{x_{0}^{2}+y_{0}^{2}+z_{0}^{2}}={\left(\sqrt{R_{t}^{2}+d^{2}}\pm r\right)}^{2}$$
where the ‘+’ of ‘±’ indicates the largest tangent sphere, and the ‘-’ indicates the smallest tangent sphere. The surface enveloped by the torus is apparently a sphere. Along with Figure 4, we can infer that the surface enveloped by the torus when *α* ≠ 90° is a different part of the spherical surface with the same radius as that when *α* = 90°. Therefore, among the wheel parameters, the envelope radius is only related to the parameters *d*, *R_t_*, and *r*.□

According to the aforementioned analysis, when there is no wheel error, the radius of the HSR can be expressed by the parameters of the grinding wheel as follows:(9)$$R_{g}=\sqrt{d^{2}+R_{t}^{2}}\pm r$$
where the ‘+’ of ‘±’ indicates the outer spherical surface and ‘-’ indicates the inner spherical surface.

The contact line between the wheel and spherical surface is a complete arc. The projection on the *X_w_Z_w_* plane is shown in Figure 5. The contact line is the chord of the workpiece circle on the *X_w_Z_w_* plane, and its length is *L_c_.* The intersection of the mid-vertical line of the contact chord and the workpiece axis is the position of the center of the sphere *O_w_*, and the distance of the chord center from the workpiece origin is *d_c_*. The relationship between the radius of the sphere to be processed and the parameters of the contact line is:(10)$$R_{g}=\sqrt{{\left(\frac{L_{c}}{2}\right)}^{2}+d_{c}^{2}}$$

Equation (10) demonstrates that the radius and position of the contact chord ultimately determine the dimensions of the spherical surface, which provides a basis for the subsequent tool adjustment.

The aforementioned indicates the geometric principles of the HSR grinding tool under ideal conditions. However, in actual processing, the rotation accuracy of the grinding wheel axis and spindle, straightness and positioning accuracy of the feed axis, and tool setting error of the wheel lead to the failure of these principles. As shown in Figure 5, we first establish the workpiece *X_w_Y_w_Z_w_* coordinate system: the workpiece rotation axis is indicated by the *Z_w_* direction, the *Y_w_* direction is determined according to the common vertical line *O_w_O_t_* of the tool rotation axis and workpiece rotation axis, the pedal on the *Z_w_* axis is the workpiece coordinate origin *O_w_*, and the direction of the *X_w_* axis is determined according to the right-hand rule. *X_m_Y_m_Z_m_* is the coordinate system of the machine tool. It is advisable to make the *X_m_Y_m_Z_m_* origin coincide with *O_w_*, the *Z_m_* axis coincides with the workpiece rotation axis, and the angle between the *X_m_* axis and *X_w_* is *φ*.

The machining experiment was conducted on an *XZCB* four-axis ultra-precision machine tool, as shown in Figure 6. The radial rotation accuracy of the workpiece spindle was less than 0.015 μm, the radial rotation accuracy of the grinding spindle was less than 1 μm, and the B-axis resolution was used to control the *α* angle of the grinding axis to be less than 0.005 arcseconds. The positioning resolution of the grinding spindle along the X-and Z-axis was less than 1 nm. Under this condition, we can ignore the error of the machine tool body and the rotation error of the wheel spindle. The main error sources are as follows: (1) the radius error of the grinding wheel, namely, the radius error of the wheel torus *δ_r_* and the pitch radius error *δ_Rt_*; and (2) the four-term tool pose and positional errors of the grinding wheel *(δ_o_*, *δ_α_*, *δ_d_*, *δ_z_*), as shown in Figure 5, where the vertical distance error δ_O_ is caused by the non-intersection of the grinding wheel spindle *Z_t_* and the workpiece spindle *Z_w_*, the angle error *δ_α_* of the two axes, the distance error *δ_d_* between the center point of the grinding wheel torus and *O_w_*, and the positional error ensuring the inner and outer sphere concentric *δ_z_* of *Z_m_*. When the values of the four errors are fixed, the wheel still has the following two degrees of freedom (and does not affect the spherical shape): the rotation of the wheel and the angle *φ* between the workpiece coordinate system and the machine tool coordinate system. When *φ =* 0, the workpiece coordinate system *X_w_Y_w_Z_w_* and machine tool coordinate system *X_m_Y_m_Z_m_* coincide. As shown in Figure 3, *φ* represents the revolution angle of the tool spindle around the workpiece spindle, which has the same effect as the workpiece rotation; therefore, the value of *φ* does not affect the machining error of the HSR. Note, the value of *φ* may affect the parallelism between the tool spindle and the *X_w_Z_w_* plane. According to the calculation formula of the angle between the tool spindle *Z_t_* axis and the *X_w_Z_w_* plane, the relationship between the two can be obtained as follows:(11)sinΔβ=sinφ·sinα
where Δ*β* is the angle between the *Z_t_* axis and *X_w_Z_w_* plane. To make the actual tool adjustment effective, it is necessary to ensure that the overlapping of the machine tool coordinate system and workpiece coordinate system is maximized; that is, *φ* must be sufficiently small. Because |Δ*β*| < 0.005° can easily be guaranteed in the actual assembly process, and usually *α* > 15° during HSR processing, that is, |sin*α*| > 0.26, thus *φ* < 0.02° can be obtained according to Equation (11). The coincidence of the coordinate systems *X_m_Y_m_Z_m_* and *X_w_Y_w_Z_w_* can be guaranteed, the direction of *δ_O_* can be assumed to coincide with the *Y_m_* axis, and *d* and *α* are on the *X_m_Z_m_* plane. We usually adjust the wheel in the machine tool coordinate system in practical machining; therefore, the coincidence of the coordinate systems *X_m_Y_m_Z_m_* and *X_w_Y_w_Z_w_* is necessary. Under this condition, *δ_O_* can be regarded as the *Y_m_*-directional height alignment error of the wheel on the machine tool. According to the analysis in Section 2.1, the angle error *δ_α_*, which does not affect the radius and shape of the sphere, is not discussed herein.

According to the aforementioned analysis and Theorem 1, among the main error sources, the error that affects the spherical shape is the height error *δ_O_* produced by the non-intersection between *Z_t_* and *Z_w_*. There is a functional relationship between the deviation of the machined surface shape, spherical surface, and *δ_O_*. We can use a homogeneous coordinate transformation to transform the points of the wheel torus from the *X_t_Y_t_Z_t_* coordinate system to *X_m_Y_m_Z_m_*, and the translation matrix is as follows:(12)$$T=T_{1}\cdot T_{1}=\left[\begin{array}{cccc} 1 & & & \\ & 1 & & \\ & & 1 & d\\ & & & 1 \end{array}\right]\left[\begin{array}{cccc} 1 & & & \delta_{O}\\ & 1 & & \\ & & 1 & \\ & & & 1 \end{array}\right]$$
where *T_1_* and *T_2_* represent translational transformations along the *X* and *Y* axes, respectively. The rotational matrix is:(13)$$R=\left[\begin{array}{cccc} 1 & & & \\ & \cos\alpha & -\sin\alpha & \\ & \sin\alpha & \cos\alpha & \\ & & & 1 \end{array}\right]$$The coordinates of any point of the wheel torus in the *X_m_Y_m_Z_m_* coordinate system are (*x, y, z*), and the coordinates of this point in the grinding wheel coordinate system are (*x*_1_, *y*_1_, *z*_1_). From Equations (12) and (13), we obtain the following:(14)$$\left[\begin{array}{c} x\\ y\\ z\\ 1 \end{array}\right]=R\cdot T\left[\begin{array}{c} x_{1}\\ y_{1}\\ z_{1}\\ 1 \end{array}\right]=\left[\begin{array}{c} x_{1}+\delta_{O}\\ y_{1}\cdot \cos\alpha -z_{1}\cdot \sin\alpha \\ y_{1}\cdot \sin\alpha +z_{1}\cdot \cos\alpha +d\\ 1 \end{array}\right]$$Substituting Equation (14) into Equation (1) yields the torus in the coordinate system *X_m_Y_m_Z_m_* as follows:(15)$${\left(\sqrt{\left(x+\delta_{O}\right)+\left(y\cdot \cos\alpha -z\cdot \cos\alpha \right)}-R_{t}\right)}^{2}+{\left(y\cdot \cos\alpha +z\cdot \cos\alpha +d\right)}^{2}=r_{t}^{2}$$According to Equation (2), we can obtain the domain
(16)$$R_{t}-r\leq \sqrt{{\left(x-\delta_{O}\right)}^{2}+{\left(y\cdot \cos\alpha +z\cdot \sin\alpha -d\cdot \sin\alpha \right)}^{2}}\leq R_{t}+r$$According to the principle of the wheel envelope, the envelope surface is rotationally symmetrical about the *Z_w_* axis. For any point (*x*_3_, *y*_3_, *z*_3_) on the envelope surface, the distance *L* to the *Z_w_* axis satisfies the following:(17)$$L_{z,max}^{2}={\left.\max\left(x^{2}+y^{2}\right)\right|}_{z=z_{3}}$$
or
(18)$$L_{z,min}^{2}={\left.\min\left(x^{2}+y^{2}\right)\right|}_{z=z_{3}}$$
where *L_z, max_* indicates the distance between the point on the outer sphere and the *Z_w_* axis, and *L_z, min_* indicates the distance between the point on the inner sphere and the *Z_w_* axis. There is a question of whether spherical contour in the presence of a height error *δ_o_* is equivalent to an optimization problem such as Equation (17) or Equation (18) under nonlinear constraints, indicated by Equations (15) and (16). The contour point coordinates (*z*_3_*, L_z_, θ*) were obtained in the cylindrical coordinate system, which were converted to the Cartesian coordinate system (*x*_3_, *y*_3_, *z*_3_) via Equation (19) as follows:(19)$$\left\{\begin{array}{l} x_{3}=L_{z}\cdot \cos\theta \\ y_{3}=L_{z}\cdot \cos\theta \\ z_{3}=z_{3} \end{array}\right.$$We can fit the contour points (*x*_3_, *y*_3_, *z*_3_) under *δ_o_* to the spherical surface using the least squares method and calculate the corresponding sphericity error.

Considering the inner spherical surface as an example, the nominal radius of the inner spherical surface is 10.000 mm, and without the loss of generality, the corresponding wheel parameters are as follows: *d* = 5.196 mm, *α* = 35.264°, *r* = 1.000 mm, and *R_t_* = 7.350 mm. The actual sphere is an incomplete hemisphere, and the height error *δ_O_* causes the theoretical sphericity to vary for different hemisphere ranges. Considering the lip of the HSR as the zero point in the Z direction, as shown in Figure 3, the sphericity error results of various spherical ranges in the Z direction were simulated as shown Figure 8. Under the same height error *δ_O_*, the larger the calculated spherical range, the larger the sphericity. Under the same calculated spherical range, the sphericity error was nearly linear with the height error *δ_O_*, and the linearity error was less than 1%. Therefore, the height error *δ_O_* can be reversed according to the measured sphericity of the processed inner spherical surface, allowing the height adjustment of the wheel to align with the *Z_w_* axis. Note, the range of the measured spherical surface needs to be consistent with the range of the simulated spherical surface.

The main errors that affect the spherical form and positional accuracy were as follows: (1) the shape error of the grinding wheel; namely, the radius error of the wheel torus *δ_r_* and the pitch radius error *δ_Rt_*, and (2) the grinding wheel pose and positional errors (*δ_o_*, *δ_d_*, *δ_z_*). The radius of the sphere is affected by the *δ_r_*, *δ_Rt_*, and *δ_d_.* The sphericity is affected by the *δ_o_.* The position of the sphere affects *δ_z_*. All these errors reflected on the workpiece by the wheel can be compensated by on-machine measurements.

### 2.2. Analysis of Tool Wear on HSR Machining

According to the geometric analysis of generating spherical grinding with a cup wheel as indicated in Section 2.1, the entire spherical surface of the HSR simultaneously participates in grinding; therefore, the geometric effect of the wheel wear is reflected on the entire spherical surface. Figure 7 presents the *X_w_Z_w_* plane view of the geometric inner-grinding model. The dotted circle indicates the nominal spherical surface processed without wear, and the solid-lined circle indicates the spherical surface processed with wear. Wheel wear causes the geometrical contact line *AB* to change to the contact torus *A_1_A_2_-B_1_B_2_*. According to the characteristics of the two rotary axes enveloping a sphere with wheel wear, for any chord *A_i_B_i_* selected on the section of the contact torus *A*_1_*A*_2_-*B*_1_*B*_2_, the chord length *A_i_B_i_* and the distance *d_i_* from the chord center point *E_i_* to *O_w_* both determine the unique spherical radius *R_g_*. Without loss of generality, we consider the contact line as *A*_1_*A*_2_ with wheel wear. Namely, the wear causes contact line *AB* to become contact line *A*_1_*A*_2_; *L_c_* and *d_c_* have changed in Equation (8), which causes the spherical radius error. By obtaining the derivative of *d_c_* on the right-hand side of Equation (10), we obtain the following:(20)$$\frac{\partial R_{g}}{\partial d_{c}}=\frac{1}{\sqrt{{\left(\frac{L_{c}}{2d_{c}}\right)}^{2}+1}}$$

Equation (20) demonstrates that *d_c_* is approximately proportional to the radius of the sphere in a small range of *d_c_*. The changes in *L_c_* and *d_c_* caused by wear are significantly small; thus, we can substitute the theoretical values into the right-hand side of Equation (20). In this condition, the spherical radius can be controlled by adjusting the *d_c_* value according to the coefficient calculated by using Equation (20).

The wear of the inner spherical machining causes the actual feed in the workpiece axis to become smaller, thereby causing an error in the spherical center in the workpiece axis, as shown in Figure 7a. The wear of the outer spherical machining causes a change in the feed of the wheel along the wheel axis and does not change the position of the spherical center in the workpiece axis.

Namely, grinding wheel wear causes the spherical radius and positional errors by affecting the actual feed, *L_c_,* and *d_c_* during wheel processing. If these spherical surface errors can be measured on-machine, we can compensate for the errors in the wheel wear.

### 2.3. Compensation Strategy and Process

According to the analysis in Section 2.1 and Section 2.2, the adjustment of the wheel in the machine tool coordinate system can be divided into the following three steps:Adjust *δ_o_* to make the surface shape an approximate sphere.Adjust the feed *F_z_* of the wheel in the *Z_m_* direction to ensure the concentricity of the inner and outer spherical surfaces.Adjust the wheel position parameter *d* to make the machining spherical radius meet the requirements.

The adjustment of the inner and outer spherical processing consists of these three steps. The initial installation error of the wheel spindle in the machine tool is adjusted in Step 1; the form of the sphere affects the radius of the spherical surface and the position of the center of the sphere. To ensure the rigidity of the inner spherical surface, it is first processed before the outer spherical surface. Inner sphere machining ensures the concentricity of the two spheres. Based on the analysis in Section 2.2, the wear of the outer sphere wheel does not affect the position of the center of the sphere. Therefore, compensating for the concentricity error occurs prior to the radius error.

We developed an on-machine measurement system for HSR grinding that measures the sphericity, concentricity, and radius of the inner and outer spheres of the HSR, as shown in Figure 6. After each machining, the B-axis is rotated; thus, the probe changes to a fixed measurement position to perform on-machine measurements of the inner and outer spherical surfaces of the HSR. The positioning accuracy of the B-axis is 0.005 arcsec, We use an inductive lever probe for on-machine measurements, and the single-point measurement repeatability of the measurement system is within 0.04 μm. The spherical sphericity measurement accuracy of the measurement system after calibration with standard balls is within 0.2 μm. The measurement accuracy of the inner and outer spherical radius was 0.5 μm, the sphericity was 1 μm, and the concentricity was 2 μm. Thus, the parameter values δ_o_ and δ_z_ can be obtained for compensation, as reflected by the measurement results of the form and positional errors of the workpiece.

## 3. Results and Discussion

### 3.1. Compensation of Height Error δ_o_

According to the compensation sequence summarized in Section 2.3, it is first necessary to compensate for the height error *δ_o_* of the wheel to minimize the sphericity of the hemisphere. We measured the sphericity errors of the hemisphere before and after compensation to determine the compensation effect using on-machine measurements (OMM). Then, we adjusted the height of the wheel using an adjustment stage with a resolution of 1 μm and measured the increments of the adjustment height with an inductive probe. 

According to the simulation results shown in Figure 8, the larger the measurement range, the more significantly the sphericity reflects on the height error. The measurement trajectory of the spherical surface is shown in Figure 9. Owing to the limitation of the spatial interference of the HSR support rod, the spherical range in the *Z_w_* direction was obtained as 7 mm. The spherical generation method has inherent advantages for the rotational symmetry of HSR, and the measurement result of the spherical circumferential roundness is within 0.3 μm. Therefore, we chose to decrease the sampling density in the spherical circumferential direction to improve the measurement efficiency. We measured three meridians at the average C-axis angle and 30 points for each meridian. Finally, these points were fitted to evaluate the sphericity. The radial runout error of the electric spindle is less than one micron. The grinding wheel rotates at 8000 rpm, the workpiece rotates at 79 rpm, and the workpiece material is fused silica. The mesh number of the roughing grinding wheel is #370, and the mesh number of the finishing grinding wheel is #2000. The model and configuration of the spindle of the two wheels used for processing the inner and outer spherical surfaces were the same. therefore, there was no difference between the adjustment methods of the two grinding wheels. We did not distinguish between the compensation results of the two spindles.

Table 1 demonstrates the five selected results of the sphericity measurement and the height error prediction before and after compensating for the height error of the wheel. The differential thread of the tool holder provides height adjustment, and the inductive probe provided the indication of the adjusted value. The adjustment resolution is 0.1 μm. Ten experiments were conducted. After each compensation experiment, the height of the wheel was adjusted to a larger height error without the loss of generality of the compensation method. The sphericity errors within approximately 20 μm were reduced to less than 2 μm by one compensation. According to the on-machine measurement results, the sphericity error after compensation was controlled within 1.5 μm, and the corresponding height error was controlled within 2 μm.

The experiment verified that the sphericity can accurately reflect the height error level of the wheel, and after compensation processing, the sphericity can be controlled within 2 μm. The measurement results demonstrate that the meridians of different C-axis angles are similar; therefore, without loss of generality, we chose the form error of one of the meridians to observe the change reflecting the sphericity error, as shown in Figure 10. The overall shapes of the meridian errors before and after compensation were similar, indicating that a height error remained. Based on the results, the sphericity does not significantly improve after the second compensation. From the compensated results shown in Figure 10b, a regular error fluctuation was observed, which is dominant in the compensated error, and it affects the final compensation. In addition, we observed the 0.3 μm tool pattern in the circumferential direction of the spherical surface using the Talor Hobson’s roundness meter, as shown in Figure 11. Therefore, the residual sphericity error of approximately 0.8 μm after compensation may be caused by the waviness of the spherical surface. Despite increasing the density of the measurement points, the unevenness of the surface cannot be ignored.

### 3.2. Compensation of Inner and Outer Spherical Radius and Concentricity Error

We compensated for the radius and concentricity errors of the inner and outer spherical surfaces after compensating for the sphericity. Table 2 lists the measurement results before and after compensation for the inner and outer spherical radii and concentricity errors. The nominal radii of the inner and outer spheres were 10 mm and 10.7 mm, respectively. The experimental results demonstrate that following the OMM and compensation, the radius error within 70 μm can be controlled within 4 μm, and the concentricity can be controlled within 3 μm. If the radius error before compensation is within 20 μm, the radius error after compensation can be controlled within 1 μm.

The experiment verified the compensation effect of the radius error and the concentricity of the inner and outer spherical surfaces. Following compensation, we found that the radius of the inner spherical surface was almost always smaller than the nominal value, whereas the radius of the outer spherical surface was almost always larger than the nominal value. The larger the radius error before compensation, the larger the residual error after compensation. This indicates that the wheel was worn during the compensation process. The significant residual error caused by wear can be improved by secondary compensation, or it can be solved by including the statistical value of wear during the compensation process in the next compensation process. The compensation effect of concentricity is nearly unrelated to the concentricity error before compensation, which confirms our inference from the analysis presented in Section 2.2. The concentricity is compensated by adjusting the position of the wheel on the *Z_w_* axis (Figure 5); thus, the wear of the external spherical grinding wheel does not affect the position of the wheel on the *Z_w_* axis.

## 4. Conclusions

By analyzing the influence of the tool setting and wear on the radius and formation error of the HSR, a compensation strategy was proposed to reduce the grinding error of the inner and outer spherical surfaces of hemispherical shell resonators (HSR). The following can be concluded based on the analysis: (1) Sphericity can be entirely compensated for by adjusting the height error of the wheel. (2) The wheel wear of the spherical-generating grinding only affects the radius, but not the coaxiality and sphericity of the HSR. (3) There is a deterministic relationship between the wheel pose and size, form, and positional errors of the HSR. Therefore, following the on-machine measurement of the errors, we compensate for these errors by adjusting the wheel pose with a deterministic relationship. The experimental results demonstrate that the compensation method can control the sphericity of the HSR to 1.5 μm, radius error to 1 μm, and concentricity to 3 μm. This method saves time in the following polishing process. The experimental results indicate that the unevenness of the spherical surface, such as the tool pattern, affects the compensation effect of the sphericity. This unevenness requires the average of several measurement points, which reduces the measurement efficiency. We will improve this unevenness by adjusting the process parameters, and the wear caused by the compensation process will also be considered in the future.

## Figures and Tables

**Figure 1 micromachines-13-01535-f001:**
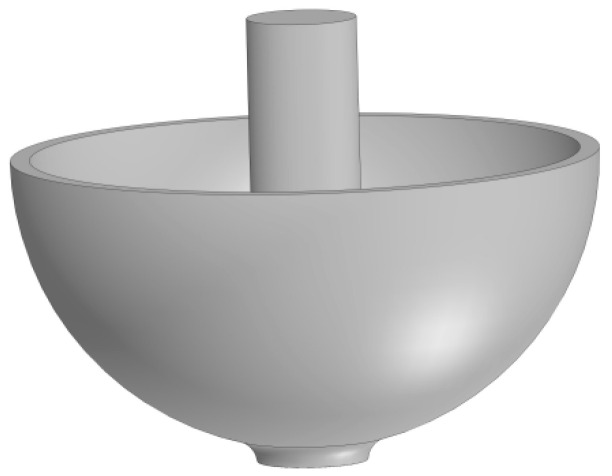
Structure of a hemispherical shell resonator.

**Figure 2 micromachines-13-01535-f002:**
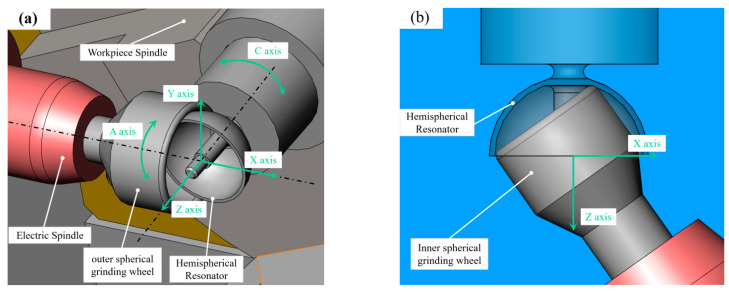
Tool-workpiece junction geometric model of HSR grinding: (**a**) external spherical grinding, (**b**) internal spherical grinding.

**Figure 3 micromachines-13-01535-f003:**
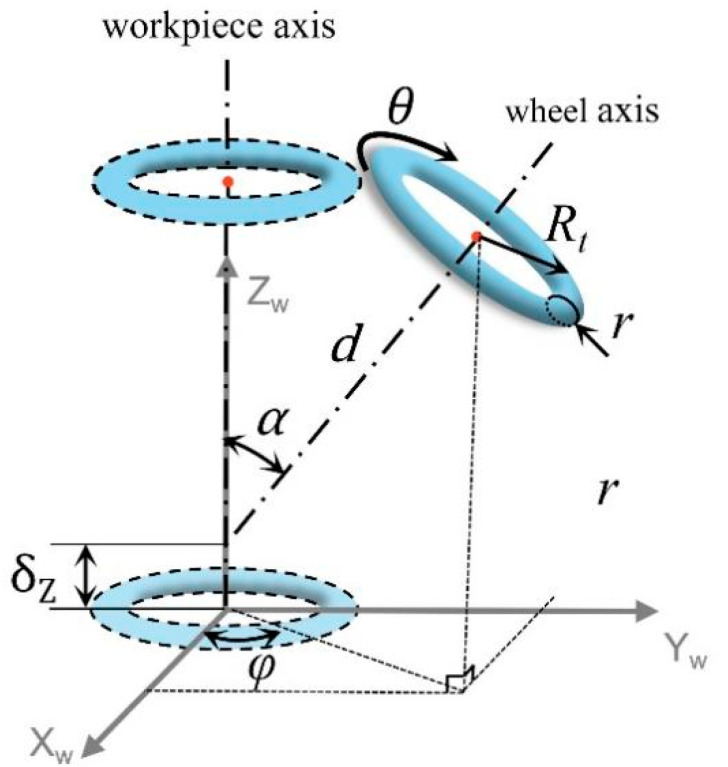
Schematic diagram of the torus.

**Figure 4 micromachines-13-01535-f004:**
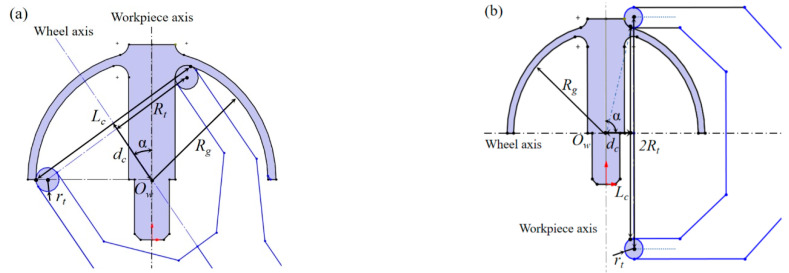
Tool-workpiece junction geometric model of HSR grinding: (**a**) external spherical grinding, (**b**) internal spherical grinding.

**Figure 5 micromachines-13-01535-f005:**
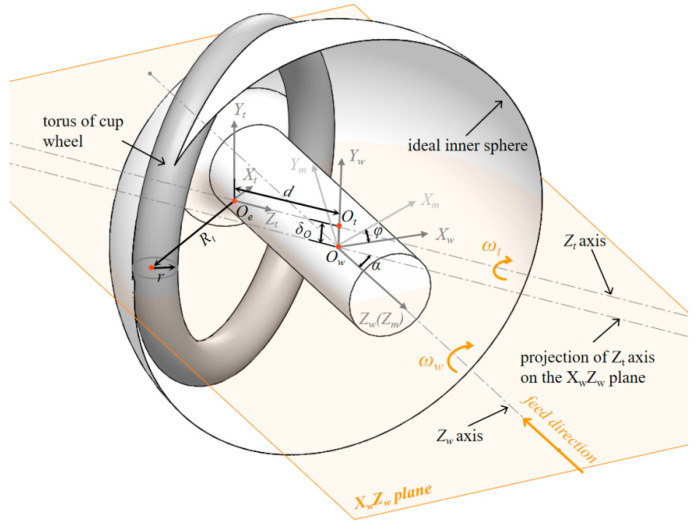
Geometrical model for HSR grinding in the presence of tool setting errors.

**Figure 6 micromachines-13-01535-f006:**
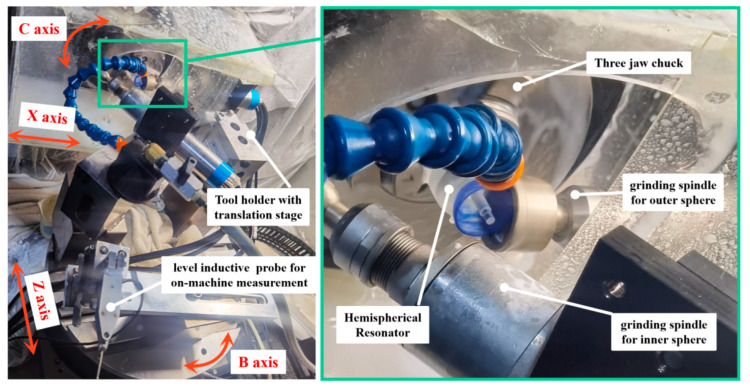
Hemispherical shell resonator grinding and on-machine measurement machine.

**Figure 7 micromachines-13-01535-f007:**
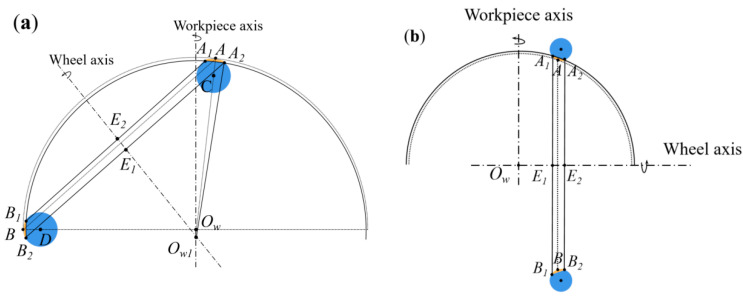
Geometric model of grinding wheel wear: (**a**) inner sphere machining wear; (**b**) outer sphere machining wear.

**Figure 8 micromachines-13-01535-f008:**
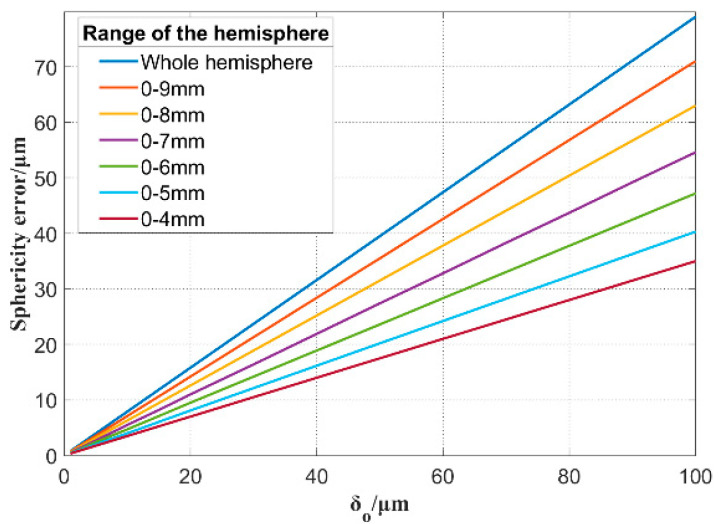
Sphericity error of different ranges of the hemisphere caused by height error.

**Figure 9 micromachines-13-01535-f009:**
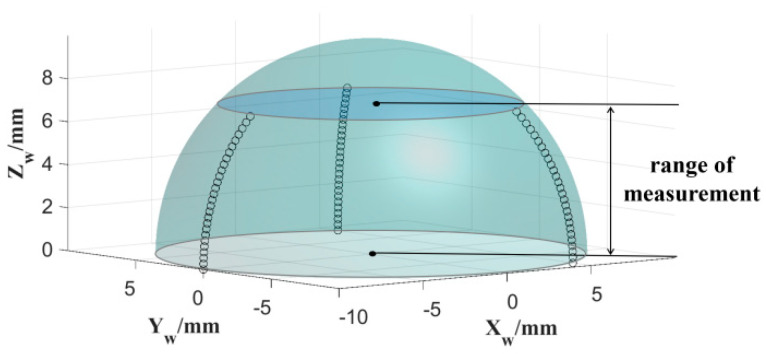
Measurement track of the hemisphere.

**Figure 10 micromachines-13-01535-f010:**
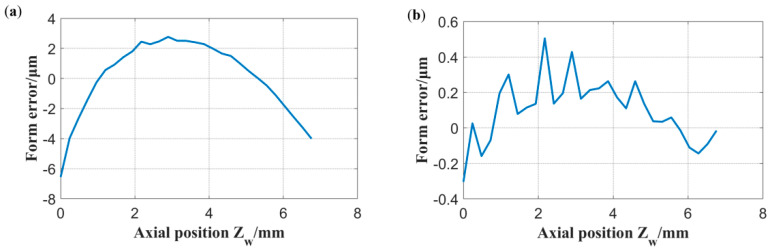
Formation error of a meridian: (**a**) initial error, (**b**) residual error after compensation.

**Figure 11 micromachines-13-01535-f011:**
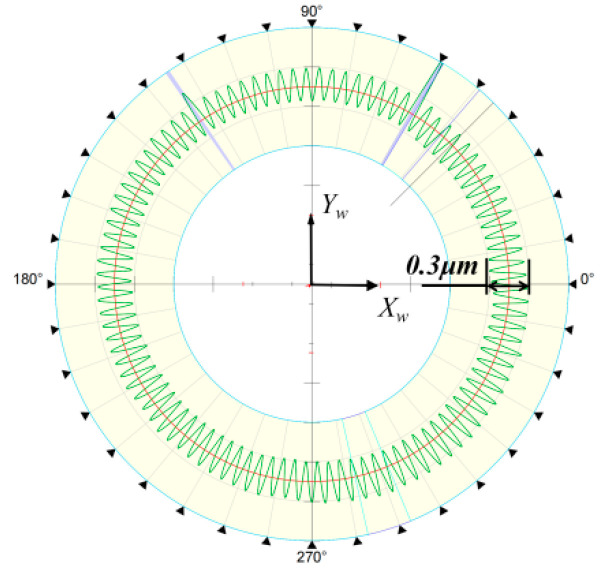
Circumferential tool pattern of the workpiece.

**Table 1 micromachines-13-01535-t001:** Results before and after compensation of height error.

Test No.	Sphericity before Compensation/μm	Corresponding δ_o_ /μm	Sphericity after Compensation/μm	Corresponding Residual δ_o_ /μm
1	4.2	7.8	0.7	1.3
2	6.3	11.5	0.9	1.6
3	10.5	19.2	0.8	1.5
4	14.9	27.2	1.2	2.0
5	20.5	37.4	1.2	2.0

**Table 2 micromachines-13-01535-t002:** Results before and after compensation of radius and concentricity errors.

Test No.	Radius of Inner Sphere/mm		Radius of Outer Sphere/mm		Concentricity/μm	
	Before	After	Before	After	Before	After
1	9.9903	9.9997	10.7338	10.7010	14.7	0.6
2	9.9883	9.9992	10.7582	10.7019	62.5	1.6
3	9.9765	9.9990	10.7321	10.7009	52.6	1.7
4	9.9558	9.9978	10.7142	10.7006	38.5	0.8
5	9.9386	9.9958	10.7685	10.7022	76.2	2.4

## Data Availability

The data presented in this study are available upon request from the corresponding author. The data is not publicly available because it is part of an ongoing study.

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
