# Peer review of "Geometric Error Analysis and Compensation in Spherical Generating Grinding of Hemispherical Shell Resonators"

_micromachines, 2022, doi:10.3390/mi13091535_

Round 1

Reviewer 1 Report

The article presents a method for compensating for shape and dimensional errors of ground spherical hemispherical shell resonator surfaces. Mentioned errors are important from the point of view of the performance of the hemispherical resonator gyroscopes.

I view the entire article very positively. The analytical considerations are clearly presented. The compensation procedure is also well described. The conclusions presented are adequate to the previously presented content.

My comments, apart from the first one, are mainly of an editorial nature:

1.    In the part devoted to experimental research, it would be advisable to specify technological parameters - grinding parameters, workpiece material, designation of grinding wheel (or its characteristics), etc.

2.    A lot of symbols appear in the article - it would be worthwhile to supplement the article with a list of symbols to improve the readability.

3.    Some references to literature and drawings, are not indicated. From figure 4 onwards, the numbering also does not match.

4.    In figure 2 (page 6, rows180), "grinding spindle for inner sphere" occurs twice.

5.    In rows 203-207, the designation for tool spindle Zw appears. It is rather Zt. Zw does not form an angle with the plane XwZw.

6.    In equation 12 there is T1*T1 - the indices at T should be different (because they represent different matrices); these symbols should be explained.

7.    On page 8, row 262 "...the larger the sphericity value". - shouldn't it be "the larger the sphericity error"?

8.    In line 273 - the sentence should not start with a symbol.

9.    line 291 - instead of equation 8 it should be equation 10.

10.  The figure on page 10 should be larger to make the symbols readable.

11.  Rows 332-335: it is not entirely clear which errors are meant, measurement errors (as written) or errors of the part without compensation (as the logic of the text implies).

12.  The word "unevenness" used in line 378 and in conclusion - does it refer to surface roughness and waviness?

Reviewer 2 Report

Geometry accuracy of hemispherical shell resonators(HSR) are key requirements in manufacturing, influencing the performance of resonant gyro. The tool-setting and tool wear error in grinding cause geometric errors of inner and outer sphere of hemispherical shell resonators. This manuscript proposed a compensation methods of wheel adjustments with on-machine measurement , simultaneously reducing the radius error, sphericity and concentricity of inner and outer sphere. The experiments regarding the inner and outer spherical machining, on-machine measurement, and compensation were conducted to verify the compensation accuracy and effect.I recommend this manuscript to be published in Micromachines if the following issues are troubleshot.

1. The description of the compensation process is insufficient. Section 2.3 presents the compensation strategy and process, and mainly describes the compensation sequence. However, what tool is used to adjust the height error, and what is the adjustment accuracy?

2. The meaning of control radius, sphericity and concentricity should be specified. From the perspective of the performance of the HSR, the significance of this study can be more prominent.

3. The method of on-machine measurement should be described in more detail so that the reader can repeat the experimental results, e.g., what accuracy and repeatability is the  on-machine measurement system? Has it been verified? Probe model?

4. The font size of the pictures should be uniform. e.g., Figure 5 and Figure 8.
